# Exploring barriers for trachomatous trichiasis surgery implementation in gamo zone, Southern Ethiopia

**Chuchu Churko** [1] *, **Mekuria Asnakew Asfaw** [1], **Zerihun Zerdo** [2]

**1** Collaborative Research and Training Center for Neglected Tropical Diseases, College of Medicine and Health Sciences, Arba Minch University, Arba Minch, Ethiopia, **2** Department of Medical Laboratory Science, College of Medicine and Health Sciences, Arba Minch University, Arba Minch, Ethiopia

* churkochuchu2005@gmail.com

## Abstract

**Data Availability Statement:** All relevant data are within the manuscript.

**Funding:** This research was fully funded by Arba Minch University with project code of GOV/AMU/TH-NTD/CRTC/01/12. The website of the University

### Background

Trachomatous trichiasis is the leading infectious cause of blindness worldwide. The World Health Organization recommends eyelid surgery to reduce the risk of visual impairment from trichiasis. Unfortunately, the number of cases operated has grown less than expected. An understanding of barriers is fundamental for instituting measures to increase surgical uptake. Therefore, the aim of this study was to explore barriers of TT surgery implementation.

### Methods

A qualitative study design was employed in December 2019. Purposive sampling technique was used to select three districts from Gamo zone, Southern Ethiopia. We conducted 9 FGDs and 12 in-depth interviews. Data was collected by audio tape recorder in Amharic and Gamogna languages and then transcribed to English language. The recorded interviews and focus group discussions were transcribed to verbatim (written text) and thematic analysis was done manually and reported accordingly.

### Findings

we explored a number of barriers that hindered implementation of trichiasis surgery. The recurrence of trichiasis after surgery was the main challenges faced by operated individuals. The other barriers reported are negative perception towards trichiasis surgery, lack of logistic and supplies, transportation access problem for remote communities, inadequate trained health professional, less commitment from higher officials, lack of interest of integrated eye care workers due to incentive issues, believes of patients waiting supernatural power for healing service and carelessness of patients to undertake operation.

is www.amu.edu.et. The funder had no role in the study design, data collection and analysis, role in the decision to publish, or preparation of the manuscript.

**Competing interests:** The authors have declared that no competing interests exist.

**Abbreviations:** AMU, Arba Minch University; CO, Corneal Opacity; FGDs, Focus Group Discussion; HEW, Health Extension Workers; IECWs, Integrated Eye Care Workers; IRB, Institute Research Review Board; KII, Key Informant Interview; MDA, Mass Drug Administration; NTDs, Neglected Tropical Diseases; SAFE, Surgery, Antibiotic, Facial cleanliness, Environmental hygiene; SNNPR, Southern Nation Nationality Region; TF, Trachomatous Follicular Trachoma; TI, Trachomatous Intense Trachoma; TS, Trachomatous Scaring; TT, Trachomatous Trichiasis; WHO, World Health Organization.

## Conclusion and recommendation

Post-surgical trichiasis, lack of commitment from government officials and negative perception of patients towards the disease were considered as the reported barriers for implementation of trachomatous trichiasis. Closely supervising the integrated eye care workers would be the first task for district health offices to increase the uptake and improve the quality of service. Logistics and supplies should be made available and adequate to address all affected people in the community.

## Author summary

Despite the scale-up of surgical services to eliminate blinding trachoma, the current surgical activity is not effectively tackling the backlog. There are limited studies done previously that explore barriers on implementation of trachomatous trichiasis surgery in Ethiopia. Therefore, understanding barriers is fundamental for instituting measures to increase surgical uptake. Hence, reliable population-based data on barriers towards trichiasis surgery implementation is very crucial for planning effective trachoma control programs, for the country like Ethiopia where trachoma ranks the first in the list of high burden countries. Our finding showed that post-surgical trichiasis, lack of commitment from government officials and negative perception of patients towards the disease were considered as the reported barriers for implementation of trachomatous trichiasis. Closely supervising the integrated eye care workers would be the first task for district health offices to increase the uptake and improve the quality of service. Logistics and supplies should be made available and adequate to address all affected people in the community.

## Introduction

Trachoma is a preventable and treatable disease caused by *Chlamydia trachomatis*. It is the leading infectious cause of blindness in the world affecting mostly the poorest community [1]. The World Health Organization simplified classification of trachoma infection as Follicular trachoma (TF), inflammatory trachoma (TI), trachomatous scaring (TS), Trachomatous Trichiasis (TT) and corneal opacity (CO). TF is described as (of at least 0.5 mm) in the upper tarsal conjunctiva; Pronounced inflammatory thickening of the tarsal. TI is when conjunctiva that obscures more than half of the deep normal tarsal vessels. TS is described as the presence of scarring in the tarsal conjunctiva and CO means the presence of easily visible corneal opacity which obscures at least some of the pupil [2].

Trachomatous trichiasis (TT) is defined as in-turning of eyelashes to the eye globes and repeated touching of the eyelashes leads to scaring of the cornea which end up to visual impairment/permanent blindness. It remains the leading infectious cause of blindness worldwide [3]. Trachomatous trichiasis is a consequence of progressive conjunctival scarring caused by recurrent infection with *Chlamydia trachomatis*. It causes painful corneal abrasion, introduces infection, and alters the ocular surface, eventually leading to irreversible blindness from corneal opacification [4].

The World Health Organization (WHO) has launched an initiative to eliminate blinding trachoma by the year 2020 using the SAFE strategy: Surgery, Antibiotic, Facial Cleanliness and Environmental improvement [5]. Surgical correction of the upper eyelid using tarsal rotation

procedure is the most effective intervention for trichiasis. Trachomatous Trichiasis surgery treatment is provided free or subsidized in most trachoma endemic settings including Ethiopia [6].

Trachoma remains a problem in the poorest societies of the world. As of 2014, an estimated 21 million people were afflicted with active trachoma, 7.3 million of whom have trichiasis and 2.2 million of whom are either completely blind or severely visually impaired. The majority of this blinding trachoma occurs in African countries [7]. According to the WHO weekly epidemiological record 2019, 142.2 million people live in trachoma endemic areas and 2.5 million people requires urgent surgery to trachomatous Trichiasis, the late stage of blinding trachoma [8].

In 2017, 70 million people lived in trachoma endemic areas in Ethiopia; that's 44% of the global burden of active trachoma [9]. Trachoma is the second most common cause of blindness in Ethiopia exceeded only by trachoma where trichiasis affects 3% of people above 14 years of age [10]. According to the national survey conducted in 2006, trachoma accounts for 11.5% of all blindness and 7.7% of people with low vision. It is estimated that over 138,000 people in Ethiopia are already blinded by this disease [11].

In the Southern Nations, Nationalities and People's Region (SNNPR), where ORBIS implements its rural program and trachoma control initiatives, the prevalence is even higher. Based on evidence generated from district baseline trachoma surveys in ORBIS supported rural project areas, the prevalence of active trachoma ranges from 22% to 56% and that of trachomatous trichiasis from 1.1% to 6.4% [12]. More recent evidence showed that Arba Minch zuria, Chencha and Dita districts have the highest TT backlog in Gamo zone, Southern Ethiopia [12].

In trachoma endemic countries, surgery usually is performed by non-physician health workers. There is currently a major global effort to scale up surgical services to clear the current trichiasis backlog by 2020, with more than 200,000 surgeries being performed annually [13].

However, surgical provision has generally been insufficient [14]. To address this, many countries, including Ethiopia, have made considerable efforts to scale-up surgical services in recent years. Unfortunately, despite this increased provision, the number of cases operated has grown less than expected. This is due to a range of service and patient-specific barriers.

The impact assessment in 2016 showed that 33 districts had TT prevalence below the elimination threshold level of 0.1% among the total population when compared to 2014 report. 276 districts had prevalence in between 0.1% and 0.9%, 357 districts had prevalence between 1% and 4.9% and 32 districts had above 5% prevalence of trichiasis. The survey showed that about 665 districts in Ethiopia needs TT service surgery [15]. As of 2017 report, 29 districts have prevalence of above 5%; 357 district between 1 to 4.9%; 304 districts have TT prevalence of 0.2% to 0.9%; and 44 districts have prevalence of below 0.2% with almost no difference with 2016 report [16].

Despite the scale-up of surgical services in recent years and one part of WHO strategy to eliminate blinding trachoma is surgery for TT patients, current surgical activity is not effectively tackling the backlog. An understanding of barriers is fundamental for instituting measures to increase surgical uptake. Federal ministry of health also recommended conducting research on why TT patients refuse to accept the surgery in Ethiopia. Therefore, the aim of this study was to explore barriers of TT surgery implantation in Gamo zone, Southern Ethiopia, 2019.

## Methods and materials

### Ethics statement

This study was conducted in accordance with the Declaration of Helsinki, and all participants' data were strictly confidential throughout the study. Ethical clearance was obtained from the

Institutional Research Ethics Review Board (IRB) of Arba Minch University, (reference number, IRB/171/12; date of approval, March 5, 2020). Verbal informed consent was secured from study participants after explaining the objective and purpose of the study. To maintain confidentiality, no personal identifiers were used on data collection forms and the recorded data were not accessed by a third person, except the principal investigators.

## Study settings

This study was conducted in Dita, Chencha and Arba Minch zuria districts, Gamo zone, Southern Ethiopia. The districts were purposively selected based of recommendation of Orbis international Ethiopia because these districts had high trichiasis backlog with estimated TT backlog of Dita (367), Chencha (327) and Arba Minch zuria (511), respectively.

Gamo zone is one of the zones in the SNNP region of Ethiopia. It is recently departed from Goffa zone, formerly known as Gamo Goffa zone. The zone is located 505 km south from Addis Ababa, having 13 districts and 4 city administrations. According to the 2007 Ethiopian central statistics agency census, the zone had a total population of 1,341,901 of which 668,230 were men and 673,671 were women. Majority of the population 1,292,653 (96.33%) live in rural area. The 2019 profile indicated that there are 5 hospitals, 33 private clinics and 53 health centers in recently formed Gamo zone.

## Study design and period

A qualitative study design was carried out from December 1 to 30, 2019 to explore barriers towards trichiasis surgery implementation in Gamo zone, Southern Ethiopia.

## Sample size calculation

Since this was qualitative study the authors did not use sample size calculation, rather the authors predetermined the number of focus group discussion (FGDs) and key informant interviews (KIIs). Nine FGDs and 12 in-depth interviews were expected to be conducted that would reach idea saturation.

## Selection of study participants and sampling procedures

Key informants were selected purposively based on their engagement in various positions of health system. District and health centre heads, neglected tropical diseases (NTD) focal persons and integrated eye care workers (IECWs) were involved in the key informant in-depth interview. On the other hand, health extension workers (those who are recruited from the community they serve, are deployed to service after 1 year formal pre-service training provided after completing 10[th] grade of education), trichiasis patient who did not undergo surgery and operated individuals were included in FGDs. District health office heads were contacted and asked to suggest clusters which they believed to have high TT cases. Then, health extension workers, cases and operated individuals were identified in each district that has high trichiasis load reported. From three districts 9 FGDs were conducted; 3 health extension worker, 3 TT patients (those individuals who had one or more lashes touching the eyeball or evidence of epilation) and 3 previously operated groups and each group comprised of 6–9 members.

## Data collection

Semi-structured key informant and FGDs interview topics were developed and pretested to increase reliability of the tool. All in-depth interview information and focus group discussions were recorded by an audio tape recorder. For every interview the interviewers were took notes

while recording. The interviews were conducted in Amharic language since it is the national language of Ethiopia and all of key informants were literate whereas the focus group discussions were conducted in Gamo language (the local language of Gamo ethnic group). The interview was conducted in comfortable environment in a private room. The duration of each KII was between 20 minutes and 40 minutes. The FGDs were conducted at their kebeles (the lowest administrative unit in Ethiopia) where the study participants are living. To reduce the risk of COVID-19 infection both interviewers and interviewees were used face mask during the data collection period. In addition, after each interview, the recorder was cleaned with sanitizer.

## Data processing and analysis

Data from FGDs and key informant interviews was transcribed, coded and analysed thematically based on the emerging themes on perception, challenges during and after operation, barriers and willingness of trichiasis patients for operation. The recorded audio in Gamo language (the local language of the study population) was transcribed in to Amharic language (the national language for Ethiopia). The transcription was checked independently by the team members for verification and for accuracy with simultaneous audio playing. After validation of the Amharic transcript, it was translated the text into the English language. The data was analysed and coded manually through thematic content analysis approach. All codes or categories were developed by the corresponding/first author.

## Data quality control

To ensure quality of data, the interview and focus group questions were well discussed by the team members before actual data collection. The interview and focus discussion questions were conducted by those individuals who had previous research work experience and first degree and above holders. Supervision was done throughout data collection process by the investigators. Intensive two day training was given for data collectors on how to conduct in-depth interviews and focus group discussion.

## Result

### Background characteristics of the participants

We conducted 9 FGDs (3 FGDs those who did not undergo TT surgery; 3 FGDs those who received TT surgery and 3 FGDs health extension workers) and 12 key informant interviews. The minimum size of FGDs was 6 and maximum number of 9 participants with age range of 27–75 years old. Twelve in-depth interviews were included in this study (NTDs focals, integrated eye care workers (IECWs), health center and districts health office heads from three districts of Gamo zone). Of the 12 in-depth interviews, three were females. The youngest and oldest age of the in-depth interview participants were 31 and 41 years old, respectively (Table 1).

### Awareness of trichiasis patient towards trachomatous trichiasis and its surgery

Majority of the trichiasis patients did not clearly know what trichiasis is and its consequences. But they explained the risk factors of their current eye problems.

**Table 1. Socio-demographic characteristics of focus group discussants.**

| Case group | | |
| --- | --- | --- |
| **Variables** | **Categories** | **Frequency (%)** |
| Age | Below 35 years old | 5 |
| | 35–45 years old | 7 |
| | Above 45 years old | 9 |
| Sex | Male | 4 |
| | Female | 17 |
| Previously operated participants | | |
| Age | Below 35 years old | 3 |
| | 35–45 years old | 8 |
| | Above 45 years old | 15 |
| Sex | Male | 6 |
| | Female | 20 |
| Health extension workers | | |
| Age | 25–35 years old | 10 |
| | 35–45 years old | 8 |
| Sex | Female | 18 |

*"I don't know what the name of the disease is and its consequences. I think it is caused by poor sanitation like living with domestic animals and open defecation. Flies touch our eyes and cause eye disease." (FGD 4, 75 years old male participant)*

*"I do not know the name of the disease but in our community if there is itching and discharge from eyes, it is called eye disease (Ayfe Hargie). It is cause by contact with dust and smokes from kitchen." (FGD8, 49 years old woman participant)*

However, few of TT patients ever heard about trichiasis. They also knew the complication if not operated for trachomatous trichiasis. A woman mentioned that

*"If an eye is presented with symptoms of inward growth of eye lashes and discharges, then it is trichiasis. A person may be blind if not treated early for trichiasis" (FGD 4, 45 years old woman).*

## Willingness of trichiasis patients to take trachomatous trichiasis surgery service

In focus group discussion, individuals did not want to undergo surgery because they need supernatural intervention to get relief from the problem. They believed that even if the surgery service is available, they were not eager to take surgical treatment rather God's healing. The discussants replied to question why not they underwent operation for their eye problem:

*"I believe that God has power to heal my eyes. I am waiting His healing service." (FGD5, 56 years old woman)*

*"I was treated for trichiasis long period ago and operated by trained health workers but the treatment/surgery gave me temporary relief for not more than one year. Now I have tearing of eyes, aching of eyes, growth of lashes inward and unable of identifying objects. That is why I did not want to use surgical service again. I believe that God will heal me in the future." (FGD5, 65 years old woman)*

On the other hand, patients from other discussants have opposed the idea of the above discussants. They complained that they repeatedly went to health facility to take surgical treatment but they were unable access the service.

*"I am willing to undergo surgery but the service is not available in the health facility when I went there. There are no supplies to give the service in the health facilities." (FGD8, 51 years old woman)*

### Awareness of previously operated patients towards trachomatous trichiasis and its surgery

On the other hand, almost all of the TT operated discussants had well known about trichiasis and its complication if not being operated. However, majority of them complained that they have operated their eyes twice or more times and the inward of eyelashes growth come again after one to two years of operation. Respondents answered to the questions why they were operated their eyes?

*"I am operated because of inward growth of lashes in to the eyes. Before operation, I had tearing and aching sensation in my eyes. After operation it was good for about one year. Then the symptoms of the problem come thereafter. I had operated two times in my life but the symptoms still exists" (FGD2, 66 years old man)*

### Challenges and improvements faced during and after surgery among previously operated individuals

The operated individuals were asked to respond for the question what challenges they faced during surgical procedure. Most of the participants raised almost similar complaints regarding challenges of TT surgery. One of them said that

*"I had pain during operation because of needle injection and suturing. After surgery, for some days I had feeling of discomfort and pain on the operation sites. But after some period it becomes subsided." (FGD1, 40 years old, woman participant)*

The other FGD discussants reported that his eyes were improved after surgery and currently doing his daily jobs. Some of the focus group discussants experienced minor pain during surgery and they were thankful and appreciated the IECWs.

*"I felt pain during injection and no pain was experienced in my eyes after operation. I was operated seven months back and currently I am in a good condition working my routine duties like other normal individuals. God bless my doctors (those health professionals who operated my eyes)."(FGD3, 69 years old man)*

### Barriers of trichiasis surgery

**Key informant interviews.** Key informants responded that during field supervision the patients were requesting when to get surgical service. One of the KIIs said that:

*"Every time we were surprized by patients asking why not is the service available to us at health facility. Even they come to our office and complaining when and where they could get the surgical service."(KI, 45 years old woman)*

Most of FGD participants and key informants tried to respond important barriers for trichiasis surgery implementation in their districts. One of them explained that interventions were undertaken to control trachoma; however there were challenges in the implementation phase. The man said that

*"Despite tremendous interventions given in our district to control and eliminate trachoma, it is still public health problem in rural kebeles. Especially trichiasis is affecting our poor community. Transfer out of trained health professionals from one health facility to another and lack of logistics and supplies are the main challenges for implementation of trichiasis surgery." (KI, NTD focal, 40 years old, male)*

*"We observed high number of TT patients at remote kebeles. Trichiasis clearance survey was not conducted in these kebeles because of remoteness. The survey included those kebeles which are easily accessible for transportation." (KI, NTDs focal, 35 years old man)*

In one of the district, key informant complained that some of components of SAFE strategy are not applied in our district.

*"Trichiasis surgery by itself is not enough to eliminate trichiasis but also behavioural change and communication personal and environmental hygiene practices would be done to reduce the burden of the disease. Government and other concerned bodies would put their efforts for application of SAFE strategies in the district." (KI, Head of health office, 43 years old man)*

The other barriers for TT surgery were as follows:

*"We are facing challenges in TT surgery because of inadequate trained health professionals. One of the IECW went for education and no one is substituted and therefore no report is coming to our office."(KI, 39 years old man)*

*"Patients have fear of pain during surgery and they believed that they might lose their sight after operation."(KI, 43 years old woman)*

Incentive issue was also considered as barrier for implementation of TT surgery. Orbis international pay for those health professionals who provided surgical treatment for patients. But from some period onward there was limited provision of incentives for IECWs.

*"We were asked one IECW why he is not operated TT patients and why didn't he report his works. He said that where is the incentive for the previous surgery? Orbis did not pay for my previous work and that is why I am not conducting TT surgery."(KI, 40 years old Woman)*

All district key informants reported that they received almost no report of TT surgery in the last year. They noted that inadequate trained health professionals, lack of supplies and materials and currently structural reformation of zone and districts in SNNP region were considered as the main challenges for TT surgery implementation.

*"I can say that in our district most of health facilities are not actively providing TT surgery, because IECWs are not interested to conduct the surgery. The district had no provision of supplies for facilitation of the surgery." (KI, 35 years old man)*

*"The current structural reformation of districts made us difficult to control trained health professionals. They are transferring out to office level and left clinical works-focus on managerial works." (KI, 45 years old man)*

There were also complains on the readiness of the districts towards implementation of trachoma in general. The disease especially trichiasis was not considered as priority problem when compared to others. It is implemented when there is campaign programs and the district health offices did not give due attention for the program-trachoma elimination.

*"Readiness of the woreda is low towards disease control. It is not considered as priority problem and the disease is neglected at district level. Attention is given only during MDA and other funds; otherwise no one look it as routine work."(KI, 43 years old woman).*

### Health extension workers

Health extension workers discussed on the question why trichiasis patients did not undergo surgery in their district:

*"In our community, some trichiasis patients are not willing to get the service when provided. They become careless to take surgery; they prefer epilation rather than surgery." (FGD6, HEW, 27 years old Woman)*

"Some of people in our district have negative perception towards trichiasis surgery. They fear that the surgery may lose their vision after surgery." (FGD9, HEW, 32 years old woman)

*"Some of patients complained recurrence of the disease even after surgery." (FGD7, HEW, 36 years old).*

*"In our community, some people have willingness to use the service; however there is poor readiness of health facilities to provide TT surgery. Lack of supplies and logistics were the major barrier mentioned by the health care providers" (FGD9, HEW, 30 years old woman).*

### Discussion

One of the goals of the United Nation is to ensure healthy lives and promote well-being for all at all ages by the year 2030. In this goal (Goal 3) sub section 3.3 clearly stated that By 2030, end the epidemics of AIDS, tuberculosis, malaria and neglected tropical diseases and combat hepatitis, water borne diseases and other communicable diseases [17]. Trachoma is one of neglected tropical diseases aimed for elimination by the year 2020 in Ethiopia. But still it is considered as public health problem in many parts of Ethiopia especially in the study area.

In this study we found that most of trichiasis patients could not identify the name of trichiasis and its consequences. When asked about the name of the eye problem and its cause, they said that it is just eye disease (ayfe hargie) and caused by frequent contact with smokes from kitchen and living with domestic animals. This finding is consistent with a qualitative study done in Kenya indicated that the study subjects who underwent operation aware that trachoma is caused by smokes from kitchen they use for cooking meals and all the FGDs believed as flies that are a result of the animals near the household also contribute to the transmission of the disease [18]. A Cochrane review also revealed that presence of domestic animals near to human dwellings should be addressed for the sake of environmental sanitation otherwise it is difficult to control trachoma [19].

The present study indicated that all trichiasis operated discussants had awareness about trachomatous trichiasis and its complication. They responded that it is disease of the poor and caused by poor personal and environmental sanitation. The possible explanation for this finding might be due to trained health workers provide health education on risk factors and prevention measures of trachoma during while providing operation service.

The finding of the current study demonstrated that most of operated trichiasis patients complained that even if they are happy in the operation provided to them and got relief from their pain, the symptoms come back after a year. The disease becomes recurrent after some period of operation. FGDs from operated group reported that they have operated more than once in their life. The other FGDs also agreed to this complain as the symptom aggravated after one to two years of operation. Many quantitative studies revealed that incidence of postoperative trichiasis is higher than WHO recommended threshold level (target a cumulative postoperative trichiasis incidence of ≤10% at one-year post-surgery) [20]. Bowman et al found recurrence rates of approximately 55% in the Gambia [21]. West et al found a 28% recurrence rate in Tanzania 2 to 8 years after surgery [22]. On the other hand, in Ethiopia studies showed that lower prevalence of postoperative trichiasis than other countries. For example, a study done in Amhara region, Ethiopia reported that postoperative trichiasis rate was 24.7% whereas another study in Ambasel district, South Wollo zone, Amhara region it was 23.8% in 2020 [23,24]. The possible explanation might be poor quality of service or it might be suggested by reinfection of patients because of lack knowledge and poor practice towards trachoma transmission and prevention measures.

The present study revealed that few trichiasis patients who did not undergo surgery had negative perception towards trachomatous trichiasis surgery. The discussants believed that they may lose their sight after surgery. They prefer epilation rather that operation and waiting supernatural intervention for healing their problem. This is true for a study conducted in Northwest Ethiopia in which the investigators did not get many cases for surgery that they complained of a bad outcome of surgery [25]. It is also consistent with study done in Ethiopia and other different areas [21,26,27]. This might indicate community awareness creation is not undertaken in the effectiveness of trichiasis surgery in the study areas. There might also be concern on the quality of surgery service because of the many operated patients complained recurrence of trichiasis.

On the contrary, some of trichiasis patients opposed the negative perception of the above discussants. The patients said that they had willingness to be operated and save their eyes from loss of vision but did not get service because of inadequate provision of the service. This finding is in line with report from Tanzania in which about 22% tried to get the service but couldn't get the surgery because of barriers on the provider's side [28]. Similarly, other study showed that some patients (4%) reported that they had in fact attended a health facility for surgery but not received treatment. It was not possible to determine the reasons why the health facility had not provided surgery [26]. The possible reason might be lack of commitment from government and other concerned bodies side.

Regarding barriers for trichiasis surgery implementation, apart from fear to surgery and negative perception on trichiasis surgery, the key informants reported many barriers towards implementation of trichiasis surgery. Lack of logistic and supplies and inadequate trained health professional were reported by all key informants. On the other hand, Habtamu et al revealed that in Ethiopia, surgeons reported multiple barriers to providing surgery including a lack of surgical supplies, conflicting work duties, little supervision and administrative support [29]. This suggests lack of commitment on the side of government officials and other concerned bodies for elimination of trachoma.

In this study the key informants reported that trichiasis cases are high in remote kebeles of the districts. Hard to reach kebeles did not given due attention for trichiasis surgery because of transportation access problem. The landscape of districts made them difficult to provide TT surgery even for outreach service. This result agrees with many different studies across the world [21,28,30]. This might be due to the fact that those trichiasis patients were unable to travel far distance by their foot since there is no accessibility of vehicles in remote kebeles and they are poor community and there might not have support/escort.

The other barriers raised were incentive issues for IECWs from Orbis international Ethiopia and carelessness of some patients to take the service if provided. This might be lack of commitment of health professionals to serve community. This might also reflect lack of commitment by the program implementer like Orbis International Ethiopia.

A limitation of this study is that pure qualitative study has been conducted in this study. We recommend identifying other barriers using strong designs to influence policy makers. We did not include zone, regional and federal health officials in this qualitative survey.

## Conclusion and recommendation

In conclusion, the most indicated barriers towards trichiasis surgery implementation were recurrence of trichiasis among operated patients, inadequate supplies and logistics, inadequate trained health professionals, accessibility problem to the service and lack of commitment among trained health professionals due to incentive issues. Closely supervising the integrated eye care workers would be the first task for district health offices to increase the uptake and improve the quality of service. In-service trainings should be given to maintain the existing system and increase the number of IECWs. Logistics and supplies should be made available and adequate to address all affected people in the community. Further research will be done for identifying predictors of trichiasis recurrence among operated patients.

## Declarations

### Ethical approval and consent to participate

Ethical approval was obtained from Arba Minch University (AMU) Institute Research Review Board (IRB), (reference number, IRB/171/12; date of approval, March 5, 2020). Support letter was obtained from the district health offices to facilitate the data collection. Informed verbal consent was obtained from each study participants before proceeding to data collection. The objective of the study was explained for each selected kebele administration. All individuals were advised about the disease complication and its prevention measures after data collection.

## Acknowledgments

We are grateful to all data collectors and supervisors, who tried their best and committed themselves to data collection. We would also like to express our appreciation to the study participants.

## Author Contributions

**Conceptualization:** Chuchu Churko.

**Data curation:** Chuchu Churko, Mekuria Asnakew Asfaw, Zerihun Zerdo.

**Formal analysis:** Chuchu Churko, Mekuria Asnakew Asfaw.

**Funding acquisition:** Zerihun Zerdo.

**Investigation:** Chuchu Churko, Mekuria Asnakew Asfaw, Zerihun Zerdo.

**Methodology:** Chuchu Churko, Mekuria Asnakew Asfaw.

**Project administration:** Chuchu Churko, Zerihun Zerdo.

**Resources:** Chuchu Churko, Zerihun Zerdo.

**Software:** Chuchu Churko, Mekuria Asnakew Asfaw.

**Supervision:** Chuchu Churko, Mekuria Asnakew Asfaw, Zerihun Zerdo.

**Validation:** Chuchu Churko, Mekuria Asnakew Asfaw, Zerihun Zerdo.

**Visualization:** Chuchu Churko, Mekuria Asnakew Asfaw, Zerihun Zerdo.

**Writing – original draft:** Chuchu Churko.

**Writing – review & editing:** Chuchu Churko, Mekuria Asnakew Asfaw, Zerihun Zerdo.

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
