## [Decision Letter · Decision Letter 0]

7 Jul 2021

Dear Mr. CHURKO,

Thank you very much for submitting your manuscript "EXPLORING BARRIERS FOR TRICHIASIS SURGERY IMPLEMENTATION IN GAMO ZONE, SOUTHERN ETHIOPIA, 2019" for consideration at PLOS Neglected Tropical Diseases. As with all papers reviewed by the journal, your manuscript was reviewed by members of the editorial board and by several independent reviewers. In light of the reviews (below this email), we would like to invite the resubmission of a significantly-revised version that takes into account the reviewers' comments. 

We cannot make any decision about publication until we have seen the revised manuscript and your response to the reviewers' comments. Your revised manuscript is also likely to be sent to reviewers for further evaluation.

Sincerely,

Michael Marks

Deputy Editor

Michael Marks

Deputy Editor

Reviewer's Responses to Questions

**Key Review Criteria Required for Acceptance?**

**Methods**

-Are the objectives of the study clearly articulated with a clear testable hypothesis stated?

-Is the study design appropriate to address the stated objectives?

-Is the population clearly described and appropriate for the hypothesis being tested?

-Is the sample size sufficient to ensure adequate power to address the hypothesis being tested?

-Were correct statistical analysis used to support conclusions?

-Are there concerns about ethical or regulatory requirements being met?

Reviewer #1: The objectives of the study are clearly stated. The study population is clearly described. The sample size is also acceptable for such a qualitative study. The FGDs and in-depth interviews were fairly transcribed and analyzed but lacks clarity due to weak translation into English language. Ethical requirements were strictly observed.

Reviewer #2: --No hypothesis but this is a qualitative study

--Study design appropriate but unclear that best practices were followed

--population not clearly described, how people were chosen is not clear

--no discussion about sample size calculations so unclear if the sample size is sufficient

**Results**

-Does the analysis presented match the analysis plan?

-Are the results clearly and completely presented?

-Are the figures (Tables, Images) of sufficient quality for clarity?

Reviewer #1: Results were not presented in accordance with the analysis plan. Results of the 3 different FGDs as categorized by their affiliations (operated TT cases, unoperated TT cases and IECWs) should have been presented separately so that the barriers for TT surgery as perceived by these 3 different groups could be clearly understood by readers of the article. Similarly, the key informant interviews should have been presented separately so that the view points expressed by service providers could be clearly understood. The authors have presented the results by thematic areas which makes it difficult for readers to clearly and succinctly identify the real barriers to TT surgery. Recurrent trichiasis appears to be a major problem in the study area but the authors didn't adequately address this critical issue. Overall, the data presented are too shallow and anecdotal. I don't think that the authors have presented strong evidence to make meaningful inferences.

Reviewer #2: --No analysis plan was given. But the pre-specified analysis plan would be a helpful addition

--Results are clear but it is not certain how representative the results are given the lack of clarity on the analysis methods

**Conclusions**

-Are the conclusions supported by the data presented?

-Are the limitations of analysis clearly described?

-Do the authors discuss how these data can be helpful to advance our understanding of the topic under study?

-Is public health relevance addressed?

Reviewer #1: The conclusions are supported by the data presented and the limitations of the study are also described. However, there is a serious problem with the English language.

Reviewer #2: --limitations can be expanded depending on the details of the analysis (see comments below)

**Editorial and Data Presentation Modifications?**

Reviewer #1: (No Response)

Reviewer #2: (No Response)

**Summary and General Comments**

Reviewer #1: The study addresses an important problem for trachoma elimination program. The method used is acceptable. However, the data and evidence presented is so weak and superficial. The data synthesis and analysis also lacks clarity and depth. The manuscript needs complete revision.

Reviewer #2: This is a qualitative research study assessing barriers to trachomatous trichiasis surgery. My main comment is that the authors have not demonstrated scientific rigor with their methods (please see detailed comments below). They provide some quotes from several people but it is unclear how they decided that these were the consensus views of the community. Other comments follow:

Abstract, background: It might be better to specify “trachomatous trichiasis” as the leading infectious cause

Abstract, findings: non-Ethiopians will not know what a “kebele” is; please substitute “communities”

Abstract, conclusion: “the main barriers for trichiasis surgery implementation were from service providers and patients’ side.” � As opposed to what?

Introduction: trachoma is not the leading cause of blindness. Cataract is. You could say “leading infectious cause”?

How was the sample size determined? Were the numbers pre-specified? Or continued until saturation was reached?

How exactly were the participants of the FGDs selected? It says trichiasis patients. From where were they recruited? From a health clinic? From a campaign?

Methods: The methods for analysis are not detailed. Specifically: (A) usually qualitative data is analyzed according to some underlying theoretical model. Did the authors consider any model when analyzing? If not, please state that the analysis was done without considering a specific framework. (B) was coding of transcripts done to classify responses into specific domains? If so, who did the coding? How many people? How did you assure that different graders reached similar conclusions in their coding (eg. Kappa). (C) what software was used? Any software specific to qualitative research (eg atlas.ti or something similar)

It would be good to more clearly state who did the key informant interviews (for example, who are “district key informants”?). Many people outside Ethiopia will not know what a health extension worker is, so please give a brief description of who they are and why we should care about their opinions.

Most readers will not know what a woreda is, please define, maybe just say “(i.e., district)”

Recommendation: what exactly is the recommendation? Are there any barriers that you would prioritize based on your research? Things that would be easy/cheap to do first, then followed by more complicated things?

PLOS authors have the option to publish the peer review history of their article (what does this mean?). If published, this will include your full peer review and any attached files.

Reviewer #1: No

Reviewer #2: No
---

## [Decision Letter · Decision Letter 1]

3 Aug 2021

Dear Mr. CHURKO,

Thank you very much for submitting your manuscript "EXPLORING BARRIERS FOR TRACHOMATOUS TRICHIASIS SURGERY IMPLEMENTATION IN GAMO ZONE, SOUTHERN ETHIOPIA" for consideration at PLOS Neglected Tropical Diseases. As with all papers reviewed by the journal, your manuscript was reviewed by members of the editorial board and by several independent reviewers. The reviewers appreciated the attention to an important topic. Based on the reviews, we are likely to accept this manuscript for publication, providing that you modify the manuscript according to the review recommendations. 

Sincerely,

Michael Marks

Deputy Editor

Michael Marks

Deputy Editor

Reviewer's Responses to Questions

**Key Review Criteria Required for Acceptance?**

**Methods**

-Are the objectives of the study clearly articulated with a clear testable hypothesis stated?

-Is the study design appropriate to address the stated objectives?

-Is the population clearly described and appropriate for the hypothesis being tested?

-Is the sample size sufficient to ensure adequate power to address the hypothesis being tested?

-Were correct statistical analysis used to support conclusions?

-Are there concerns about ethical or regulatory requirements being met?

Reviewer #1: I don't think the study design is appropriate to address the stated objectives. The study population has not been clearly described. The authors simply gathered some TT backlog estimates for some selected districts but didn't even present how many of those were operated. Although it is a qualitative study, the sample size of the FGDs, KIIs, etc. should fairly represent the study population. For example, there were only 21 operated TT cases in the FGDs. What was the total number of TT cases operated in the three districts? No data is given. It's not also clear if they have referred to the TT case registers in the health facilities. They were relying on the verbal confirmations provided by the NGO partner (Orbis) and some of the district health office staff.

Reviewer #2: Adequate

**Results**

-Does the analysis presented match the analysis plan?

-Are the results clearly and completely presented?

-Are the figures (Tables, Images) of sufficient quality for clarity?

Reviewer #1: The results and the analyses presented lack clarity and completeness. The descriptions/explanations provided are weak and not analytic. They are quoting some references in the discussion section that are not relevant to the findings of the study (things like the SDGs). The quotes (translations) from the FGD discussants are not written with good English. Instead of describing the identified barriers to TT surgery, the authors were rather dwelling more on discussing other broader components of the SAFE strategy and government policies for trachoma control.

Reviewer #2: Adequate

**Conclusions**

-Are the conclusions supported by the data presented?

-Are the limitations of analysis clearly described?

-Do the authors discuss how these data can be helpful to advance our understanding of the topic under study?

-Is public health relevance addressed?

Reviewer #1: The conclusions are, to some extent, supported by the data presented. Some of the limitations of the study are stated.

Reviewer #2: Adequate

**Editorial and Data Presentation Modifications?**

Reviewer #1: The paper needs thorough revision.

Reviewer #2: (No Response)

**Summary and General Comments**

Reviewer #1: The paper needs major revision.

Reviewer #2: The authors have addressed my comments.

PLOS authors have the option to publish the peer review history of their article (what does this mean?). If published, this will include your full peer review and any attached files.

Reviewer #1: No

Reviewer #2: No

Figure Files:

Data Requirements:

Reproducibility:

References

---

## [Decision Letter · Decision Letter 2]

1 Sep 2021

Dear Mr. CHURKO,

We are pleased to inform you that your manuscript 'EXPLORING BARRIERS FOR TRACHOMATOUS TRICHIASIS SURGERY IMPLEMENTATION IN GAMO ZONE, SOUTHERN ETHIOPIA' has been provisionally accepted for publication in PLOS Neglected Tropical Diseases.

Best regards,

Michael Marks

Deputy Editor

Michael Marks

Deputy Editor

Reviewer's Responses to Questions

**Key Review Criteria Required for Acceptance?**

**Methods**

-Are the objectives of the study clearly articulated with a clear testable hypothesis stated?

-Is the study design appropriate to address the stated objectives?

-Is the population clearly described and appropriate for the hypothesis being tested?

-Is the sample size sufficient to ensure adequate power to address the hypothesis being tested?

-Were correct statistical analysis used to support conclusions?

-Are there concerns about ethical or regulatory requirements being met?

Reviewer #1: Acceptable

Reviewer #2: Adequate

**Results**

-Does the analysis presented match the analysis plan?

-Are the results clearly and completely presented?

-Are the figures (Tables, Images) of sufficient quality for clarity?

Reviewer #1: Acceptable

Reviewer #2: Adequate

**Conclusions**

-Are the conclusions supported by the data presented?

-Are the limitations of analysis clearly described?

-Do the authors discuss how these data can be helpful to advance our understanding of the topic under study?

-Is public health relevance addressed?

Reviewer #1: Acceptable

Reviewer #2: Adequate

**Editorial and Data Presentation Modifications?**

Reviewer #1: Acceptable

Reviewer #2: (No Response)

**Summary and General Comments**

Reviewer #1: My comments have been addressed.

Reviewer #2: the authors addressed my previous comments; I continue to think the paper is worthy of dissemination.

PLOS authors have the option to publish the peer review history of their article (what does this mean?). If published, this will include your full peer review and any attached files.

Reviewer #1: No

Reviewer #2: No

---

## [Editor Report · Acceptance letter]

9 Sep 2021

Dear Mr. CHURKO,

We are delighted to inform you that your manuscript, "EXPLORING BARRIERS FOR TRACHOMATOUS TRICHIASIS SURGERY IMPLEMENTATION IN GAMO ZONE, SOUTHERN ETHIOPIA," has been formally accepted for publication in PLOS Neglected Tropical Diseases.

Best regards,

Shaden Kamhawi

co-Editor-in-Chief

Paul Brindley

co-Editor-in-Chief
